# Construction and alidation of a severity prediction model for acute pancreatitis based on CT severity index: A retrospective case-control study

Xiao Han, Mao-neng Hu [ORCID]*, Peng Ji, Yun-feng Liu

Imaging Center, Hefei Third Clinical College of Anhui Medical University (The Third People's Hospital of Hefei City), Hefei, Anhui Province, China

* hmn596@126.com

**Data Availability Statement:** All the data relevant to the current manuscript are available at https://osf.io/m9ckf/.

## Abstract

To construct and internally and externally validate a nomogram model for predicting the severity of acute pancreatitis (AP) based on the CT severity index (CTSI).A retrospective analysis of clinical data from 200 AP patients diagnosed at the Hefei Third Clinical College of Anhui Medical University from June 2019 to June 2022 was conducted. Patients were classified into non-severe acute pancreatitis (NSAP, n = 135) and severe acute pancreatitis (SAP, n = 65) based on final clinical diagnosis. Differences in CTSI, general clinical features, and laboratory indicators between the two groups were compared. The LASSO regression model was used to select variables that might affect the severity of AP, and these variables were analyzed using multivariate logistic regression. A nomogram model was constructed using R software, and its AUC value was calculated. The accuracy and practicality of the model were evaluated using calibration curves, Hosmer-Lemeshow test, and decision curve analysis (DCA), with internal validation performed using the bootstrap method. Finally, 60 AP patients treated in the same hospital from July 2022 to December 2023 were selected for external validation.LASSO regression identified CTSI, BUN, D-D, NLR, and Ascites as five predictive factors. Unconditional binary logistic regression analysis showed that CTSI (OR = 2.141, 95%CI:1.369–3.504), BUN (OR = 1.378, 95%CI:1.026–1.959), NLR (OR = 1.370, 95%CI:1.016–1.906), D-D (OR = 1.500, 95%CI:1.112–2.110), and Ascites (OR = 5.517, 95%CI:1.217–2.993) were independent factors influencing SAP. The established prediction model had a C-index of 0.962, indicating high accuracy. Calibration curves demonstrated good consistency between predicted survival rates and actual survival rates. The C-indexes for internal and external validation were 0.935 and 0.901, respectively, with calibration curves close to the ideal line.The model based on CTSI and clinical indicators can effectively predict the severity of AP, providing a scientific basis for clinical decision-making by physicians.

**Funding:** This work was supported by the Hefei Seventh Period Key Specialty Construction Project [Hefei Health Secretariat (2023) No. 72]; Hefei Medical Imaging Clinical Medical Research Center Project [Hefei Health Education (2022) No. 20]; Hefei Health and Wellness Applied Medical Research Project (HwK2023yb007).

**Competing interests:** The authors have declared that no competing interests exist.

## Introduction

Acute pancreatitis (AP) is a common gastrointestinal disorder characterized by sudden inflammation of the pancreas, with clinical manifestations ranging from mild, self-limiting symptoms to severe complications and high mortality rates. Early assessment of the severity of AP is crucial for predicting patient outcomes and formulating appropriate treatment strategies. Depending on the severity of AP, physicians can decide whether hospitalization is needed, when to initiate nutritional support, whether surgical intervention is required, and which pharmacological treatments to use. Thus, early assessment of the severity of AP can assist physicians in better managing the patient's condition and improving treatment outcomes. Currently, several scoring systems have been developed to predict the severity of AP, including Ranson's criteria, the Acute Physiology and Chronic Health Evaluation (APACHE) II score, and the Bedside Index for Severity in Acute Pancreatitis (BISAP) [1–3]. However, these methods have certain limitations in accuracy and clinical application, with current research suggesting that these scoring methods have high specificity but low sensitivity [4].

Recently, an increasing number of studies have focused on using CT enhancement scans to evaluate the severity of AP, employing the CT Severity Index (CTSI) in conjunction with clinical indicators. CTSI is a radiological scoring system based on CT scan results, designed to objectively quantify the severity of pancreatic inflammation, the presence of pancreatic necrosis, and extrapancreatic complications. Several studies have reported on the potential value of CTSI in predicting the severity and prognosis of AP [5–7]. However, the combined effect of CTSI with clinical parameters has not been sufficiently studied.

How can the integration of the CTSI with selected clinical indicators enhance the prediction accuracy for the severity of AP? This study aims to develop a novel clinical prediction model that combines CTSI and clinical indicators to enhance the accuracy and reliability of assessing the severity of AP. This model was developed using data from a large cohort of AP patients and validated in an external population to evaluate its performance in terms of discrimination, calibration, and clinical utility. This new approach has the potential to advance our understanding of the pathophysiology of AP and provide clinicians with a more comprehensive, individualized risk assessment tool, ultimately hoping to improve patient care and prognosis.

## Materials and methods

### Study population and design

A retrospective case-control study was used in the research. This study, approved by the Medical Ethics Committee of Hefei Third People's Hospital (Approval Number: 2023LLWL027), was conducted with the informed consent of all participating patients.

We retrospectively analyzed clinical data from patients diagnosed with AP, gathered from the database of the Third Clinical College of Anhui Medical University, Hefei, spanning June 2019 to June 2023. Following approval from the Institutional Ethics Committee, data retrieval and analysis took place from July to December 2023. Of these, 200 AP patients identified between June 2019 and June 2022 were used as the training set. This group included 134 males and 66 females, with an age range of 18 to 87 years and an average age of 48.0 years (SD = 16.4 years). Additionally, 60 AP patients treated at the same hospital from July 2022 to June 2023 were selected as the validation set, comprising 43 males and 17 females, with an age range of 19 to 79 years and an average age of 48.4 years (SD = 15.1 years). Inclusion criteria were: (1) age ≥18 years; (2) patients with complete clinical data, laboratory tests, and imaging diagnostic information. Exclusion criteria included: (1) pregnant and lactating patients; (2) patients with severe cardiac, liver, or renal dysfunction; (3) patients with an acute episode of chronic pancreatitis.

### Grouping methods and collection of observation indicators

According to the Revised Atlanta Classification (RAC) standards [8], the 200 AP patients were divided into 95 cases of mild acute pancreatitis, 40 cases of moderate to severe acute pancreatitis, and 65 cases of severe acute pancreatitis (SAP). Patients with mild and moderate to severe acute pancreatitis were collectively classified as non-severe acute pancreatitis (NSAP). The patients were divided into two groups: NSAP (n = 135) and SAP (n = 65). Relevant factors that might affect the severity of AP were collected through the hospital's electronic medical record system, including general clinical data (age, gender, BMI, history of diabetes, cause of onset, APACHE II score, BISAP score), initial blood test results post-onset [white blood cell count (WBC), red cell distribution width (RDW), percentage of neutrophils (NEUT%), neutrophil to lymphocyte ratio (NLR), fasting blood glucose (FBG), blood amylase (AMY), lactate dehydrogenase (LDH), blood urea nitrogen (BUN), serum albumin (ALB), blood creatinine (Cr), D-dimer (D-D), fibrinogen (Fib)], and imaging indicators (presence of pleural effusion, ascites in CT scan, and CTSI score).

### CTSI diagnostic criteria

All patients underwent a multi-slice spiral CT abdominal enhancement scan within 3 to 7 days of symptom onset. The diagnosis of the disease and CTSI scoring were conducted by two experienced radiologists. In case of disagreement, a third senior radiologist was consulted, and the final score was determined by majority rule. The CTSI scoring [9, 10] considers the inflammatory response in and around the pancreas as well as the area of pancreatic necrosis. A score of 0 is assigned if there is no inflammatory response; 2 points are given if there is inflammation in the pancreas or peripancreatic area. For pancreatic necrosis, 0 points are assigned if there is none; 2 points if the necrotic area is less than 30%; 4 points if necrosis covers 30% to 50% of the pancreas; and 6 points if more than 50% of the pancreas is necrotic. The total score ranges from 0 to 10, with higher scores indicating more severe acute pancreatitis.

### Statistical analysis

The data were analyzed using the IBM-SPSS software package, version 26.0. Normality of the data was tested using the Kolmogorov-Smirnov method, with $P \geq 0.05$ indicating a normal distribution. Quantitative data conforming to a normal distribution were expressed as mean ± standard deviation ($\bar{x} \pm s$) and analyzed using the Student's t-test for two independent samples. Non-normally distributed data were described using median [M(P25, P75)] and analyzed using the Mann-Whitney U test. Categorical data were described using proportions or ratios and analyzed using the chi-square test. LASSO regression identified predictive factors for SAP, further evaluated through binary logistic regression to determine their independent effects, presenting odds ratios (OR) with 95% confidence intervals (CI). A nomogram, created with R software (version 4.0.2), visualized the model, whose diagnostic accuracy was gauged by ROC curve, calibration curve, and Hosmer-Lemeshow test. Decision Curve Analysis (DCA) assessed clinical utility, with internal validation via bootstrap resampling. Statistical significance was set at $P < 0.05$.

## Results

### Baseline data

The detailed flowchart of this study was shown in Fig 1. There were no statistically significant differences between the NSAP and SAP groups in terms of age, BMI, history of diabetes, presence of pleural effusion, and cause of onset ($P>0.05$ for all). However, the proportion of males, FBG, AMY, BUN, Cr, D-D, Fib, RDW, WBC, NEUT%, NLR, APACHE II score, BISAP score,

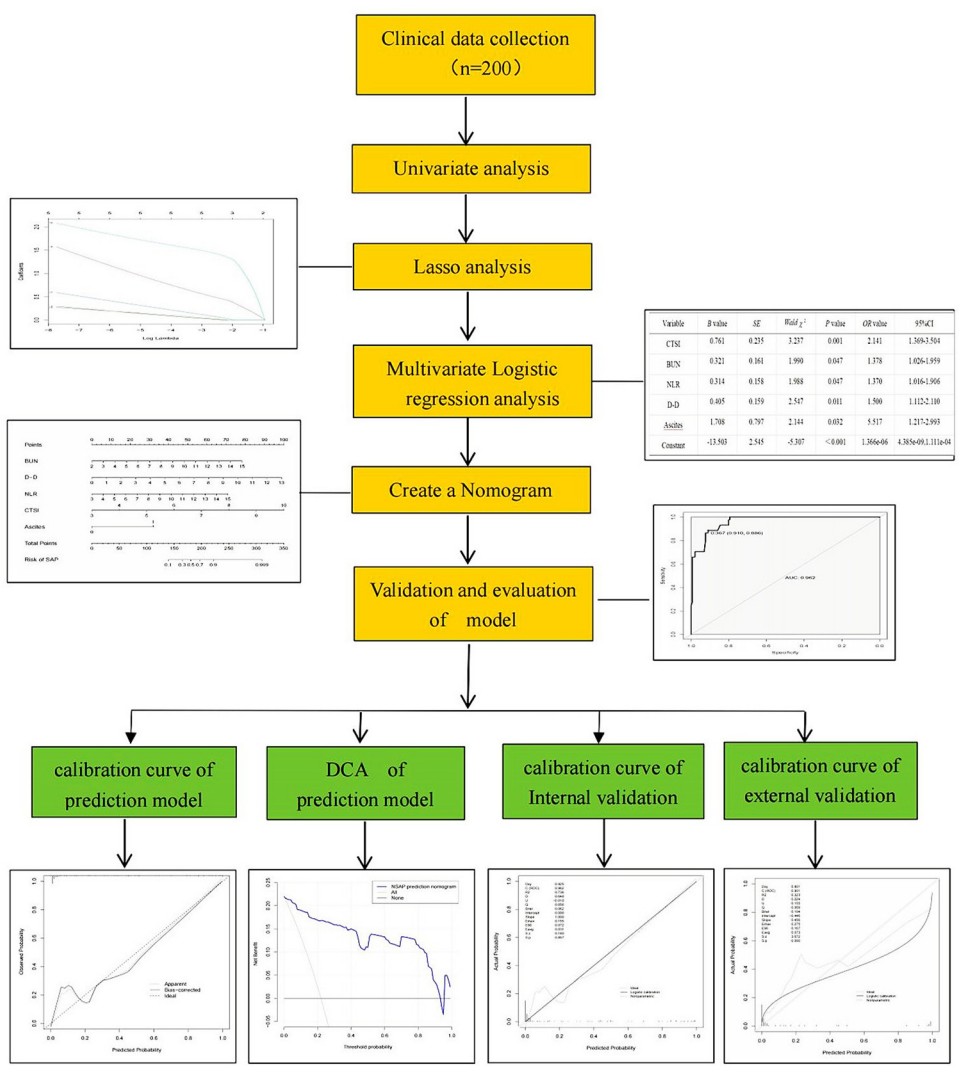

**Fig 1. Flowchart of the study process.**

and CTSI score were all higher in the SAP group compared to the NSAP group ($P<0.05$ for all). Conversely, the ALB levels were significantly lower in the SAP group ($P < 0.05$). These findings are presented in Table 1.

## Selection of predictive factors

A total of 16 factors showing differences between the SAP and NSAP groups were identified (proportion of males, FBG, AMY, ALB, BUN, Cr, D-D, Fib, RDW, WBC, NEUT, NLR, ascites, and scores of APACHE II, BISAP, and CTSI). These positive indicators were included as candidate predictive factors in the LASSO regression analysis. Through 10-fold cross-validation, a total of 5 indicators were selected as predictive factors for SAP: CTSI, NLR, BUN, D-D, and ascites (Fig 2).

## Multifactorial analysis of SAP

Using the five predictive factors identified by the LASSO regression analysis (CTSI, NLR, BUN, D-D, and Ascites) as independent variables and the occurrence of SAP as the dependent

**Table 1. Baseline information of patients in the SAP and NSAP groups.**

| Variable | SAP group (n = 65) | NSAP group(n = 135) | $Z/t/\chi^2$ value | P value |
|---|---|---|---|---|
| Age (y, $\bar{x} \pm s$) | 49.6±17.1 | 47.2±15.8 | 0.979 | 0.329 |
| Male [n(%)] | 51 (78.5) | 79 (58.5) | 7.670 | 0.006 |
| BMI (kg/m², $\bar{x} \pm s$) | 25.5±3.8 | 25.0±3.6 | 0.904 | 0.367 |
| Diabetes [n(%)] | 11 (16.9) | 26 (19.3) | 0.159 | 0.690 |
| AMY [U/L, M(P25,P75)] | 865(265, 1553) | 360(210,682) | -13.905 | <0.001 |
| FBG (mmol/L, $\bar{x} \pm s$) | 10.5±4.8 | 8.1±4.4 | 3.507 | <0.001 |
| BUN (mmol/L, $\bar{x} \pm s$) | 8.9±6.6 | 4.8±2.3 | 6.462 | <0.001 |
| Cr (mmol/L, $\bar{x} \pm s$) | 125.5±41.3 | 80.0±28.6 | 9.067 | <0.001 |
| D-D [mg/L, M(P25,P75)] | 5.2 (1.8, 8.5) | 1.3 (0.5, 2.6) | -50.215 | <0.001 |
| RDW (%, $\bar{x} \pm s$) | 18.5±7.3 | 13.5±2.7 | 7.036 | <0.001 |
| WBC (×10⁹/L, $\bar{x} \pm s$) | 14.8±5.7 | 12.1±4.4 | 3.681 | <0.001 |
| NEUT (%, $\bar{x} \pm s$) | 80.0±21.2 | 71.3±12.0 | 3.699 | <0.001 |
| NLR ($\bar{x} \pm s$) | 13.6±7.5 | 8.8±6.1 | 4.828 | <0.001 |
| Fib (g/L, $\bar{x} \pm s$) | 6.2±0.8 | 4.5±0.5 | 18.362 | <0.001 |
| ALB (g/L, $\bar{x} \pm s$) | 30.5±4.6 | 38.1±5.5 | 9.633 | <0.001 |
| APACHEII ($\bar{x} \pm s$) | 8.6±5.1 | 5.8±2.5 | 5.217 | <0.001 |
| BISAP ($\bar{x} \pm s$) | 2.1±0.7 | 1.0±0.4 | 14.110 | <0.001 |
| CTSI ($\bar{x} \pm s$) | 8.9±1.1 | 4.5±0.5 | 38.936 | <0.001 |
| Hydrothorax [n(%)] | 20 (30.8) | 40 (29.6) | 0.820 | 0.365 |
| Ascites [n(%)] | 43 (66.2) | 18 (13.3) | 57.748 | <0.001 |
| Causes of disease [n(%)] | | | 0.015 | >0.999 |
| Alcoholic | 10 (15.4) | 20 (14.8) | | |
| Biliary origin | 28 (43.1) | 58 (43.0) | | |
| Hyperlipidemia | 8 (12.3) | 17 (12.6) | | |
| Other | 19 (29.2) | 40 (29.6) | | |

variable (NSAP = 0, SAP = 1), an unconditional binary logistic regression analysis was conducted. A stepwise backward method was used to optimize the model. The results indicated that CTSI (OR = 2.141, 95%CI: 1.369–3.504), BUN (OR = 1.378, 95%CI: 1.026–1.959), NLR (OR = 1.370, 95%CI: 1.016–1.906), D-D (OR = 1.500, 95%CI: 1.112–2.110), and Ascites (OR = 5.517, 95%CI: 1.217–2.993) were all independent influencing factors for SAP ($P < 0.05$ for all) Table 2.

## Establishment of SAP prediction model

Using the independent influencing factors identified in the multifactorial analysis (CTSI, NLR, BUN, D-D, and Ascites), the regression equation for the SAP prediction model was established as follows: Logit(P) = −13.503+0.321×BUN+0.405×D-D+0.314×NLR+0.761×CTSI+1.708×Ascites. Based on this regression equation, a Nomogram was constructed (Fig 3). The area under the curve (AUC) of this model was 0.962 (95% CI: 0.939–0.987), with a sensitivity of 0.910 and specificity of 0.886 (Fig 4).

## Assessment of the accuracy of the prediction model

The Hosmer-Lemesshow test was applied to assess the accuracy of the SAP prediction model. The test revealed no statistically significant difference between the predicted incidence of SAP and the actual probability of its occurrence ($\chi^2 = 0.832$, $P = 0.660$). The Calibration curve for

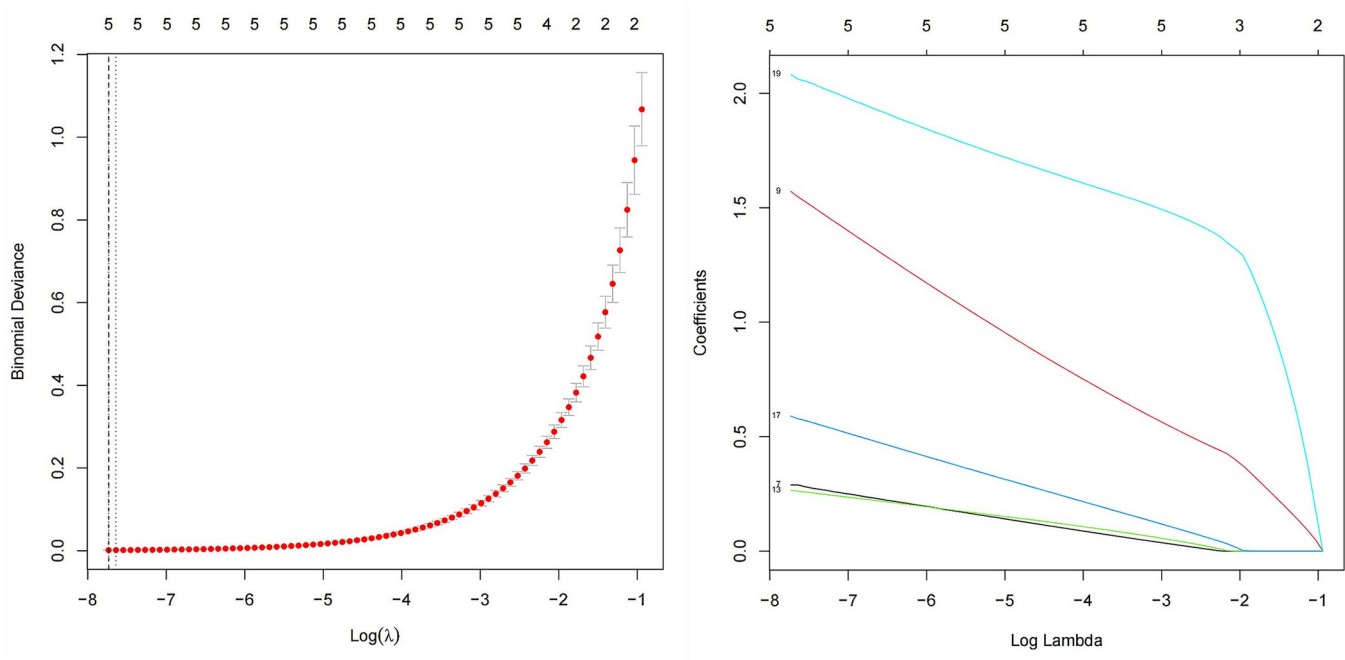

**Fig 2. Perioperative variable selection using a LASSO logistic regression model.**

this prediction model was plotted, demonstrating good consistency between the predicted risk of SAP and the actual risk, indicating high accuracy of the model. This is shown in Fig 5A.

## Clinical utility assessment of the prediction model

The clinical utility of the model was evaluated using a DCA. The DCA showed that when the predicted threshold probability of SAP occurrence ranged from 1% to 94%, the curve was consistently above the two extreme lines representing all cases as SAP and all as NSAP, as seen in Fig 5B. This indicates that the model is effective in predicting the progression of AP to SAP within a threshold probability range of 10% to 94%. In this range, appropriate treatment measures can be taken to benefit patients in clinical treatment, demonstrating good clinical utility of the model.

## Internal validation of the prediction model

The model underwent internal validation with 1000 bootstrap resamples for calibration. The results showed that the AUC value was 0.935, indicating that the model still possesses a high

**Table 2. Multi-factor logistic regression analysis of the occurrence of SAP.**

| Variable | B value | SE | Wald $\chi^2$ | P value | OR value | 95%CI |
|---|---|---|---|---|---|---|
| CTSI | 0.761 | 0.235 | 3.237 | 0.001 | 2.141 | 1.369–3.504 |
| BUN | 0.321 | 0.161 | 1.990 | 0.047 | 1.378 | 1.026–1.959 |
| NLR | 0.314 | 0.158 | 1.988 | 0.047 | 1.370 | 1.016–1.906 |
| D-D | 0.405 | 0.159 | 2.547 | 0.011 | 1.500 | 1.112–2.110 |
| Ascites | 1.708 | 0.797 | 2.144 | 0.032 | 5.517 | 1.217–2.993 |
| Constant | -13.503 | 2.545 | -5.307 | <0.001 | 1.366e-06 | 4.385e-09,1.111e-04 |

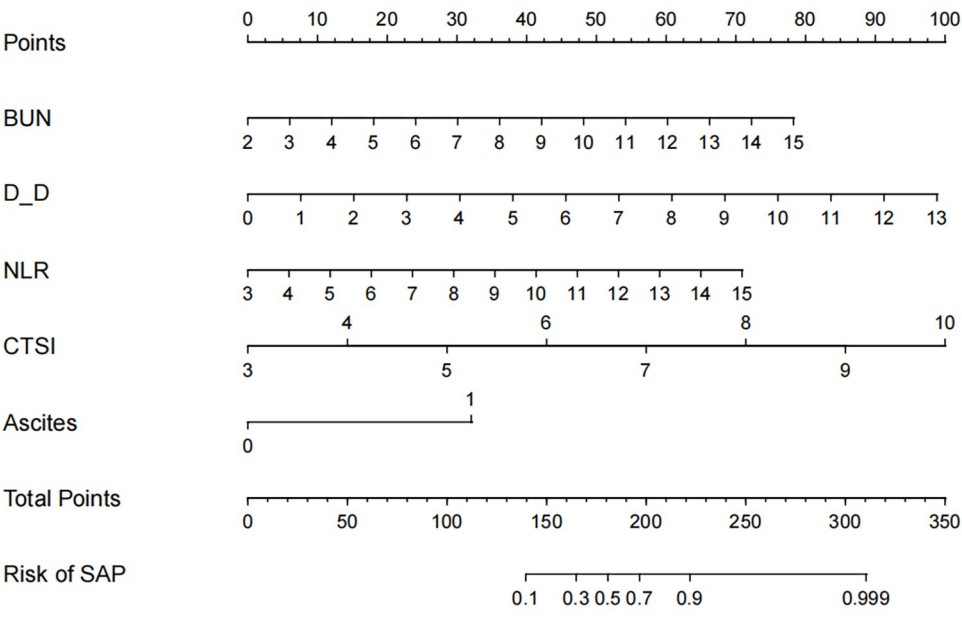

**Fig 3. Nomogram plot of SAP prediction model.**

discriminative ability. This demonstrates good consistency between the model's prediction of the severity of AP and the actual severity of AP, as seen in Fig 5C.

## External validation of the prediction model

For external validation, 60 AP patients treated at the same hospital from July 2022 to December 2023 were selected, including 43 males and 17 females; their ages ranged from 19 to 79 years, with an average age of 48.4±15.1 years. There was no significant statistical difference between the validation set and the training set in terms of age, gender, BMI, and cause of onset ($P > 0.05$ for all). The AUC of this prediction model was 0.901 (95% CI: 0.806 to 0.996) for external validation, with a sensitivity of 0.848 and specificity of 0.857. The Calibration curve plotted for this prediction model also indicated good consistency (Fig 5D).

## Discussion

Our study successfully developed and validated a predictive model combining the CTSI with clinical indicators such as BUN, NLR, D-D, and ascites. Demonstrating high accuracy and reliability, this model stands as a significant advancement in predicting the severity of acute pancreatitis. It offers clinicians a robust tool for early severity assessment, potentially leading to tailored treatment strategies and improved patient outcomes. Our findings contribute valuable insights into the integration of radiological and clinical data for enhancing acute pancreatitis management. To verify the model's effectiveness in clinical application, this study introduced DCA, an innovative approach. By setting threshold probabilities, DCA provided a reliable basis for clinical decision-making and calculated net benefits. The DCA results indicate that, within a threshold probability range of 10% to 94%, implementing personalized treatment for AP patients, as opposed to a uniform SAP treatment approach or no SAP treatment, could lead to more significant clinical benefits.

   BUN is an effective early predictor of the severity of AP, especially important in stratifying AP severity and forecasting prognosis [11]. An international study demonstrated that a BUN

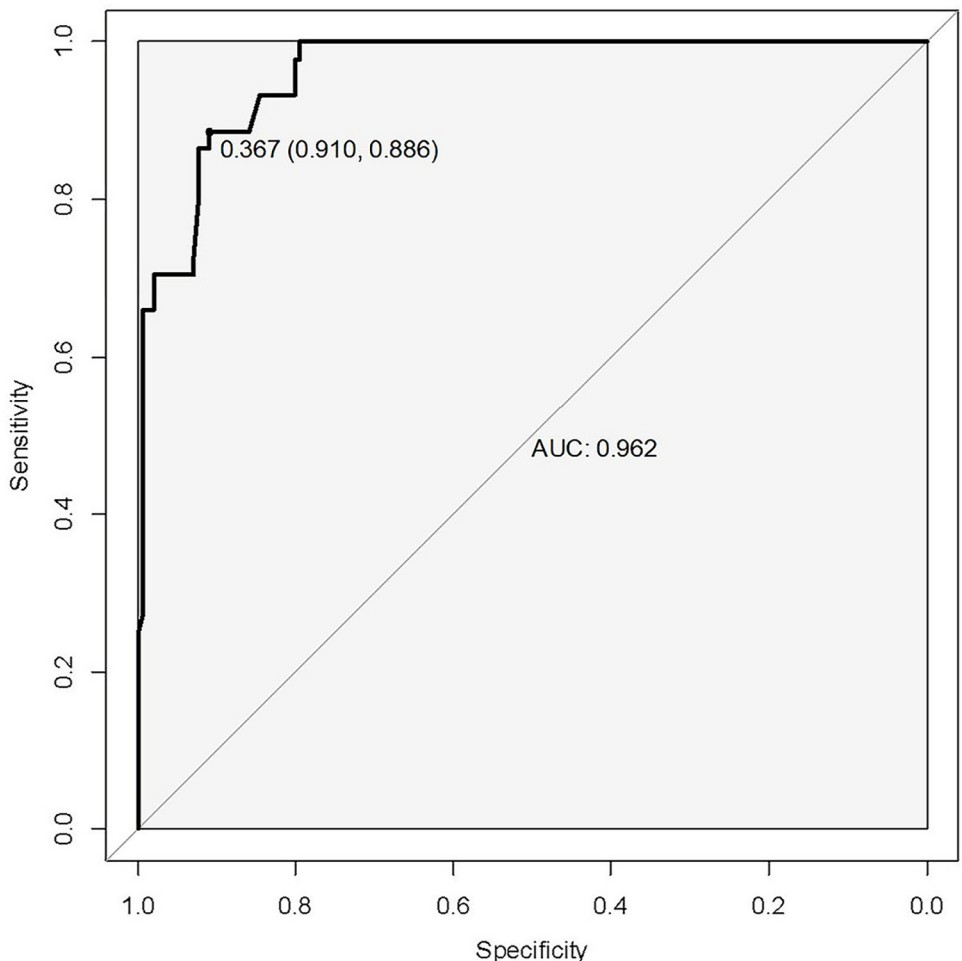

**Fig 4. Receiver operating characteristic curves for the SAP prediction model.**

level≥20 mg/dL is associated with increased mortality rates [12]. Although there is debate about the timing of BUN measurement for predicting SAP and in-hospital mortality, Lin et al. [13] found that BUN measured within 24 hours of admission has high accuracy. Research by Talukdar [14] indicated that an increase in BUN within 48 hours of admission could predict primary infective pancreatic necrosis. Koutroumpakis and colleagues [15] discovered that an increase in BUN within 24 hours correlates with accurate predictions of persistent organ failure and pancreatic necrosis, surpassing the APACHE II score. Chen's study [16] revealed that levels of PCT, CRP, HCT, and BUN within 48 hours of admission are independent risk factors for infective pancreatic necrosis, and their combination might more accurately predict secondary necrotizing pancreatitis with infection. This study confirms that BUN is an independent predictive factor for SAP; the higher the level, the greater the likelihood of SAP, consistent with previous research findings.

NLR was initially introduced as an easily measurable parameter for assessing systemic inflammation and stress in critically ill patients. A study by Abu-Elfatth [17] found that an NLR greater than 2.43 had 100% overall accuracy in predicting SAP, while the Platelet to Lymphocyte Ratio (PLR) had 87% overall accuracy at a critical point greater than 187.04. Another study suggested that the optimal cutoff value of NLR for predicting the severity of AP was 4.76,

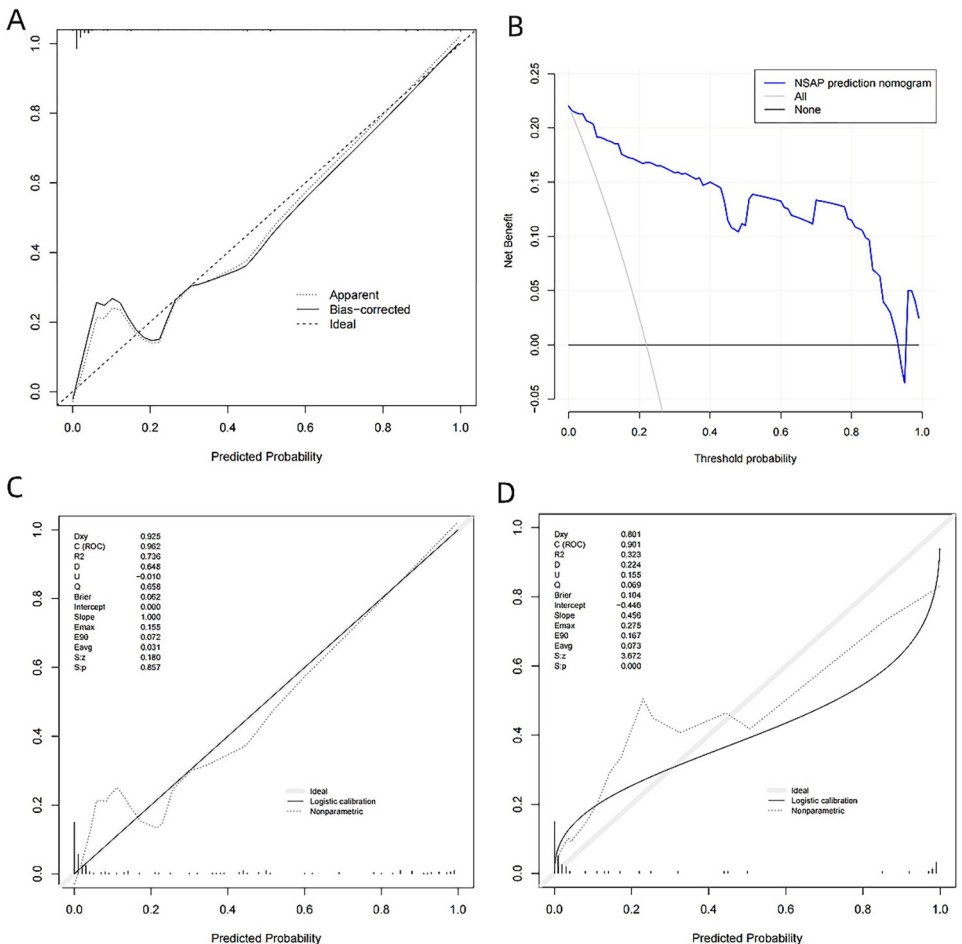

**Fig 5. Evaluation of the SAP predictive model. (A)** Calibration curve for SAP prediction model. **(B)** DAC of SAP prediction model. **(C)** Internal validation calibration curve for SAP prediction model using Bootstrap method. **(D)** Calibration curve for external validation of SAP prediction model.

and for predicting organ failure, it was 4.88 [18]. Research by Halaseh et al. [19] indicated that NLR could predict the severity of AP at initial diagnosis and might forecast adverse outcomes, the need for intensive care, and the length of hospital stay. These studies demonstrate that NLR is a useful biomarker for predicting the severity of acute pancreatitis early in the disease course. However, there is still some controversy regarding the optimal cutoff value of NLR, necessitating further research for determination.

Early mortality in the course of AP is primarily due to systemic inflammatory response. The release and activation of inflammatory cytokines lead to a hypercoagulable state in the blood, microvascular thrombosis, aggravating pancreatic tissue ischemia and hypoxia, and potentially contributing to organ dysfunction. Elevated D-dimer levels indicate increased thrombin formation and fibrinolysis, thus reflecting the formation of thrombosis in pancreatic tissue and other organs, and thereby assessing the severity of AP. Previous studies have shown that serum D-dimer levels are independently associated with the severity of AP, including a higher incidence of SAP. D-dimer is a good predictor of SAP and also a marker of endothelial dysfunction in acute pancreatitis [20, 21]. A study by Cui et al. [22] showed that D-dimer has a positive predictive value for SAP (OR = 1.21, 95%CI: 1.05–1.40). Research by Zhang et al. [23]

found that D-dimer levels could predict the severity of AP, with an average D-dimer level >7.268 indicating SAP. However, a study by Kolber et al. [24] suggested that the diagnostic accuracy of D-dimer for predicting SAP is low to moderate and not significantly different from C-reactive protein and procalcitonin. These studies indicate that D-dimer is a useful biomarker for predicting SAP, although its predictive accuracy may vary.

In the early stages of SAP, some patients develop ascites associated with AP. This phenomenon's underlying mechanism involves the occurrence of systemic inflammatory response syndrome in early SAP, leading to increased capillary permeability. Consequently, harmful substances like pancreatic enzymes and inflammatory factors leak into the abdominal cavity, resulting in acute pancreatitis-associated ascites. These substances in the ascites further exacerbate inflammation by upregulating the expression of inflammatory cytokines such as tumor necrosis factor-alpha and interleukin-6. Therefore, the presence of ascites intensifies the cytokine storm during the course of SAP. Additionally, ascites contribute to elevated intra-abdominal pressure in SAP patients, and therapeutic paracentesis to reduce this pressure is an effective means of lowering mortality [25]. Research by Yi et al. [26] found that aggressive fluid resuscitation increased ascites, a risk factor for predicting acute kidney injury in SAP. This study also indicates that the occurrence of ascites is closely related to the development of SAP, potentially leading to infections, increased intra-abdominal pressure, and other complications. Therefore, timely identification and management of ascites in SAP patients are crucial for improving prognosis.

While our model demonstrates promising accuracy in predicting the SAP, it's important to acknowledge certain limitations. First, the retrospective nature of our study may introduce selection bias and limit the generalization of our findings. Additionally, our analysis is based on data from a single center, which might not capture the variability present in a broader patient population. Although we utilized a robust statistical approach to select predictive factors and validate our model, the potential for overfitting exists, particularly given the model's reliance on a specific set of clinical indicators. Future studies, preferably multi-center and prospective in design, are needed to validate our findings and explore the integration of other potential predictors, such as genetic markers or novel biomarkers, to enhance the model's predictive capability and clinical utility.

## Conclusion

The study developed a model based on CTSI and specific clinical indicators-BUN, NLR, D-D and ascites—to predict the SAP. The results indicate that the model exhibits high accuracy and reliability in predicting the severity of AP. Notably, it demonstrated good predictive and discriminative abilities in both internal and external validations. The calibration curves showed good consistency between predicted survival rates and actual survival rates. These findings highlight the importance of integrating radiological and clinical data in the assessment of SAP, providing clinicians with a more robust tool for managing this complex disease.

## Supporting information

**S1 Checklist. STROBE statement—checklist of items that should be included in reports of observational studies.**
(DOCX)

## Author Contributions

**Conceptualization:** Mao-neng Hu, Peng Ji, Yun-feng Liu.

**Data curation:** Xiao Han, Mao-neng Hu, Peng Ji, Yun-feng Liu.

**Formal analysis:** Xiao Han, Mao-neng Hu, Peng Ji, Yun-feng Liu.

**Funding acquisition:** Xiao Han, Mao-neng Hu, Yun-feng Liu.

**Investigation:** Mao-neng Hu.

**Methodology:** Xiao Han, Mao-neng Hu.

**Project administration:** Mao-neng Hu.

**Resources:** Mao-neng Hu, Peng Ji.

**Software:** Peng Ji, Yun-feng Liu.

**Supervision:** Peng Ji, Yun-feng Liu.

**Validation:** Xiao Han, Mao-neng Hu, Peng Ji, Yun-feng Liu.

**Visualization:** Xiao Han, Mao-neng Hu, Yun-feng Liu.

**Writing – original draft:** Xiao Han, Mao-neng Hu, Yun-feng Liu.

**Writing – review & editing:** Xiao Han, Mao-neng Hu.

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
