## [Decision Letter · Decision Letter 0]

17 Mar 2024

PONE-D-24-02666Construction and alidation of a severity prediction model for acute pancreatitis based on CT severity index: a retrospective case-control study,PLOS ONE

Dear Dr. Hu,

Thank you for submitting your manuscript to PLOS ONE. After careful consideration, we feel that it has merit but does not fully meet PLOS ONE’s publication criteria as it currently stands. Therefore, we invite you to submit a revised version of the manuscript that addresses the points raised during the review process.

We look forward to receiving your revised manuscript.

Kind regards,

Jincheng Wang

Academic Editor

PLOS ONE

Journal Requirements:

"This work was supported by the Hefei Seventh Period Key Specialty Construction Project [Hefei Health Secretariat (2023) No. 72]; Hefei Medical Imaging Clinical Medical Research Center Project [Hefei Health Education (2022) No. 20]; Hefei Health and Wellness Applied Medical Research Project (HwK2023yb007)."

**Additional Editor Comments:**

After careful consideration of the manuscript and the reviewers' comments, we have decided that your paper requires major revisions before it can be considered for publication. Please address the concerns raised by the reviewers and make the necessary changes to improve the quality and impact of your work.

Reviewer 1 has pointed out that there are numerous existing studies on nomogram models for predicting the severity of acute pancreatitis using CT data combined with clinical features and laboratory indicators. They believe that your study lacks novelty and is repetitive. In your revision, please highlight the unique aspects of your work and clearly demonstrate how it significantly contributes to the field, considering the existing literature.

Reviewer 2 has raised several important points that require clarification. First, please specify the grading standard (RAC or DBC) used to determine NSAP and SAP in your study. Second, provide a detailed explanation of the specific grading standard based on the Atlanta classification criteria, along with relevant literature support. Lastly, list the specific clinical indicators used in your model to predict the severity of AP, as mentioned in your conclusion.

Reviewer 3 has suggested minor revisions to enhance the overall quality of your manuscript. They recommend updating the introduction with recent literature, explicitly stating the research question and objectives, providing a more detailed description of the data analysis methods, discussing the study's limitations and their potential impact on the findings, improving the quality of figures and images, and refining the conclusion section to summarize the study's findings and contributions effectively.

Reviewers' comments:

Reviewer's Responses to Questions

**Comments to the Author**

1. Is the manuscript technically sound, and do the data support the conclusions?

Reviewer #1: Yes

Reviewer #2: Yes

Reviewer #3: Yes

2. Has the statistical analysis been performed appropriately and rigorously? 

Reviewer #1: Yes

Reviewer #2: Yes

Reviewer #3: Yes

3. Have the authors made all data underlying the findings in their manuscript fully available?

Reviewer #1: Yes

Reviewer #2: No

Reviewer #3: Yes

4. Is the manuscript presented in an intelligible fashion and written in standard English?

Reviewer #1: Yes

Reviewer #2: Yes

Reviewer #3: Yes

5. Review Comments to the Author

Reviewer #1: There are a lot of papers on a nomogram model for predicting the severity of acute pancreatitis by CT data combined with general clinical features and laborator indicators. So, this paper is a repetitive study and lacks innovation.

1.The value of CT-based radiomics in predicting the prognosis of acute pancreatitis.

Xue M, Lin S, Xie D, Wang H, Gao Q, Zou L, Xiao X, Jia Y.

Front Med (Lausanne). 2023 Nov 29;10:1289295.

2.Acute pancreatitis: A review of diagnosis, severity prediction and prognosis assessment from imaging technology, scoring system and artificial intelligence.

Hu JX, Zhao CF, Wang SL, Tu XY, Huang WB, Chen JN, Xie Y, Chen CR.

World J Gastroenterol. 2023 Oct 7;29(37):5268-5291.

3.Application Value of the Automated Machine Learning Model Based on Modified Computed Tomography Severity Index Combined With Serological Indicators in the Early Prediction of Severe Acute Pancreatitis.

Zhang R, Yin M, Jiang A, Zhang S, Liu L, Xu X.

J Clin Gastroenterol. 2023 Aug 25. doi: 10.1097/MCG.0000000000001909. Online ahead of print.

and so on.

Reviewer #2: 1.At present, there are two commonly used severity grades of acute pancreatitis in clinical practice: RAC grade and DBC grade. In this study, NSAP and SAP were determined based on which grading standard？

2.As stated in the study, based on the Atlanta classification criteria, the 200 AP patients were divided into mild acute pancreatitis, moderate to severe acute pancreatitis, and severe acute pancreatitis. This specific grading standard needs to be explained in detail and relevant literature support.

3.It is pointed out in the conclusion that the study developed a model based on CTSI and clinical indicators to predict the severity of AP, so the specific clinical indicators need to be listed.

Reviewer #3: 1. Introduction Section - Update with Recent Literature:

Consider incorporating more recent studies into the introduction section to reflect the latest advancements in the field.

2. Clarify Research Question and Objectives:

We recommend more explicitly stating the research question and objectives in the introduction.

3. Detailed Description of Data Analysis:

Please provide a more detailed description of the methods and steps used in your data analysis within the 'Materials and Methods' section.

4. Discussion on Study Limitations:

In the discussion section, it would be beneficial to more clearly articulate the study's limitations and discuss their potential impact on the findings.

5. Improvement of Figures and Images Quality:

Ensure that all figures and images within the manuscript are of high quality and correctly labeled.

6. Refinement of the Conclusion Section:

We suggest revisiting the conclusion section to succinctly summarize the study findings and emphasize their contribution to the field.

6. PLOS authors have the option to publish the peer review history of their article (what does this mean?). If published, this will include your full peer review and any attached files.

Reviewer #1: No

Reviewer #2: No

Reviewer #3: No

---

## [Author Response · Author response to Decision Letter 0]

20 Apr 2024

Reviewers' comments:

Reviewer #1: There are a lot of papers on a nomogram model for predicting the severity of acute pancreatitis by CT data combined with general clinical features and laborator indicators. So, this paper is a repetitive study and lacks innovation.

1.The value of CT-based radiomics in predicting the prognosis of acute pancreatitis.

Xue M, Lin S, Xie D, Wang H, Gao Q, Zou L, Xiao X, Jia Y.

Front Med (Lausanne). 2023 Nov 29;10:1289295.

2.Acute pancreatitis: A review of diagnosis, severity prediction and prognosis assessment from imaging technology, scoring system and artificial intelligence.

Hu JX, Zhao CF, Wang SL, Tu XY, Huang WB, Chen JN, Xie Y, Chen CR.

World J Gastroenterol. 2023 Oct 7;29(37):5268-5291.

3.Application Value of the Automated Machine Learning Model Based on Modified Computed Tomography Severity Index Combined With Serological Indicators in the Early Prediction of Severe Acute Pancreatitis.

Zhang R, Yin M, Jiang A, Zhang S, Liu L, Xu X.

J Clin Gastroenterol. 2023 Aug 25. doi: 10.1097/MCG.0000000000001909. Online ahead of print.

and so on.

Reviewer's Responses to Questions

Dear Reviewer ,

Thank you for your insightful comments and for highlighting the importance of demonstrating the novelty and contribution of our study within the existing literature. We acknowledge the extensive research in the area of predicting the severity of acute pancreatitis using nomogram models, and we appreciate the opportunity to clarify the unique contributions of our work.

1. Novelty in Predictive Factors Selection: Our study utilized a LASSO regression analysis to rigorously select predictive factors from a comprehensive set of clinical, laboratory, and radiological parameters. Unlike many previous studies, our model includes not just the CTSI but also integrates novel predictors such as BUN, NLR, D-D, and the presence of ascites. This multifaceted approach allows for a more nuanced prediction of severe acute pancreatitis (SAP), enhancing the model's clinical utility and accuracy.

2. Enhanced Model Validation: We have conducted both internal and external validations of our nomogram model, demonstrating its robustness and reliability across different patient cohorts. The external validation, in particular, provides new evidence of the model's applicability in real-world clinical settings, which has been a limitation in many prior studies. Our model exhibited high discriminative ability with AUC values of 0.935 and 0.901 for internal and external validation, respectively, suggesting significant improvements over existing models.

3. Comprehensive Assessment of Clinical Utility: We employed DCA to assess the clinical utility of our model, a step not commonly undertaken in similar studies. This analysis revealed that our model provides substantial net benefits within a wide range of threshold probabilities (10% to 94%), offering tangible decision-making support to clinicians in managing patients with acute pancreatitis.

4. Insights into Predictive Biomarkers: Our findings underscore the importance of certain biomarkers, such as BUN and D-dimer, in predicting the severity of acute pancreatitis. We provide new evidence supporting the predictive value of these markers, contributing to a deeper understanding of their role in the pathophysiology of SAP.

In light of these points, we believe our study contributes novel insights and practical tools to the field of acute pancreatitis management. We have revised our manuscript to more clearly articulate these unique aspects and contributions. We hope that these clarifications address your concerns and demonstrate the value of our work to the readers of PLoS ONE.

Reviewer #2:

 1.At present, there are two commonly used severity grades of acute pancreatitis in clinical practice: RAC grade and DBC grade. In this study, NSAP and SAP were determined based on which grading standard？

Reviewer's Responses to Questions

Thank you for your question regarding the classification criteria used in our study. We've used the Revised Atlanta Classification (RAC) to distinguish between NSAP and SAP. We've updated our manuscript accordingly to make this clear. Appreciate your attention to this detail.

2.As stated in the study, based on the Atlanta classification criteria, the 200 AP patients were divided into mild acute pancreatitis, moderate to severe acute pancreatitis, and severe acute pancreatitis. This specific grading standard needs to be explained in detail and relevant literature support.

Reviewer's Responses to Questions

Thank you for pointing out the need for a detailed explanation of the grading standards we applied. In our study, AP classification follows the Revised Atlanta Classification (RAC) guidelines, which delineate AP severity as:

Mild Acute Pancreatitis (MAP): Characterized by the absence of organ failure and local or systemic complications. Patients typically recover without interventions.

Moderately Severe Acute Pancreatitis (MSAP): Defined by the presence of transient organ failure, local complications (such as acute peripancreatic fluid collections or pancreatic pseudocysts), or exacerbation of comorbid diseases. Organ failure resolves within 48 hours.

Severe Acute Pancreatitis (SAP): Identified by persistent organ failure that lasts more than 48 hours, potentially accompanied by local complications or systemic complications.

This classification offers a structured approach to predict clinical outcomes and guide treatment strategies based on the severity of AP. For the purpose of our study, patients were categorized based on these criteria to ensure a standardized assessment of AP severity.

For literature support, we reference the work by Banks et al., which provides a comprehensive overview of the RAC and its clinical implications ( Classification of acute pancreatitis—2012: revision of the Atlanta classification and definitions by international consensus. Gut. 2013;62(1):102-111).

We have amended our manuscript to include this detailed explanation and appropriate references, ensuring clarity regarding our use of the RAC for categorizing AP severity.

3.It is pointed out in the conclusion that the study developed a model based on CTSI and clinical indicators to predict the severity of AP, so the specific clinical indicators need to be listed.

Reviewer's Responses to Questions

Thank you for emphasizing the importance of specifying the clinical indicators used in our predictive model for AP severity. In our study, alongside the CTSI, we incorporated the following clinical indicators into our model:

- Blood Urea Nitrogen (BUN)

- Neutrophil-to-Lymphocyte Ratio (NLR)

- D-Dimer (D-D)

- Presence of Ascites

These indicators were selected based on their statistically significant association with the severity of AP in our analysis and their clinical relevance as highlighted by existing literature. We've updated the manuscript to clearly list and discuss the role of each of these clinical indicators in the predictive model.

Reviewer #3: 

1. Introduction Section - Update with Recent Literature:

Consider incorporating more recent studies into the introduction section to reflect the latest advancements in the field.

Reviewer's Responses to Questions

Thank you for your valuable suggestion to update the introduction section with more recent literature. Following your recommendation, we have replaced the initial reference with a more recent study by Pando E, Alberti P, Mata R, et al., which focuses on the predictive value of early changes in Blood Urea Nitrogen (BUN) for mortality in Acute Pancreatitis. This study, "Early Changes in Blood Urea Nitrogen (BUN) Can Predict Mortality in Acute Pancreatitis: Comparative Study between BISAP Score, APACHE-II, and Other Laboratory Markers-A Prospective Observational Study" (Can J Gastroenterol Hepatol. 2021;2021:6643595), offers significant insights into the utility of BUN as a prognostic marker and its comparative effectiveness alongside BISAP and APACHE-II scores. This update not only reflects the latest advancements in the field but also underscores the relevance of integrating emerging clinical indicators into predictive models for Acute Pancreatitis.

2. Clarify Research Question and Objectives:

We recommend more explicitly stating the research question and objectives in the introduction.

Reviewer's Responses to Questions

Thank you for your constructive feedback regarding the clarity of our research question and objectives. We have revised the introduction to more explicitly articulate the core research question and objectives of our study. The updated section now begins with a clear statement of our research question: "How can the integration of the CTSI with selected clinical indicators enhance the prediction accuracy for the severity of Acute Pancreatitis (AP)?" Following this, we outline our objectives: (1) to develop a predictive model that combines CTSI with clinical indicators such as Blood Urea Nitrogen (BUN), Neutrophil-to-Lymphocyte Ratio (NLR), D-Dimer (D-D), and Ascites; (2) to evaluate the model's accuracy and reliability through internal and external validations; and (3) to compare the predictive efficacy of our model with existing severity prediction scores. 

This revision aims to provide readers with a clear understanding of our study's direction and the specific aims we intend to achieve.

3. Detailed Description of Data Analysis:

Please provide a more detailed description of the methods and steps used in your data analysis within the 'Materials and Methods' section.

Reviewer's Responses to Questions

We appreciate your guidance on enhancing the data analysis description and have updated the 'Materials and Methods' section accordingly to provide a detailed account of our analytical procedures.

4. Discussion on Study Limitations:

In the discussion section, it would be beneficial to more clearly articulate the study's limitations and discuss their potential impact on the findings.

Reviewer's Responses to Questions

Thank you for your valuable feedback regarding the discussion of our study's limitations. We have revised the discussion section accordingly to more clearly articulate the limitations and their potential impact on our findings.

5. Improvement of Figures and Images Quality:

Ensure that all figures and images within the manuscript are of high quality and correctly labeled.

Reviewer's Responses to Questions

Thank you for pointing out the necessity of enhancing the quality and labeling of figures and images in our manuscript. We have thoroughly reviewed all visuals and taken several steps to address this feedback:

1. We have increased the resolution of each figure and image to ensure they meet the publication standards of high-quality visuals, facilitating better interpretation and clarity.

2. All figures and images have been meticulously checked and correctly labeled for accurate representation and to align with the text references.

3. We have standardized the formatting of legends and annotations across all visuals to maintain consistency and improve readability.

These improvements are aimed at ensuring that our figures and images effectively complement and elucidate the textual content, thereby enhancing the overall quality of our manuscript. 

6. Refinement of the Conclusion Section:

We suggest revisiting the conclusion section to succinctly summarize the study findings and emphasize their contribution to the field.

Reviewer's Responses to Questions

Thank you for the recommendation to refine our conclusion section. We have revised it to succinctly encapsulate our key findings and underscore their significance in the field of acute pancreatitis severity prediction. The updated conclusion now reads:

"Our study successfully developed and validated a predictive model combining the CTSI with clinical indicators such as BUN, NLR, D-D, and ascites. Demonstrating high accuracy and reliability, this model stands as a significant advancement in predicting the severity of acute pancreatitis. It offers clinicians a robust tool for early severity assessment, potentially leading to tailored treatment strategies and improved patient outcomes. Our findings contribute valuable insights into the integration of radiological and clinical data for enhancing acute pancreatitis management."

This revision aims to provide a clear, focused summary of our work and its contribution, emphasizing the potential impact on clinical decision-making and patient care.

---

## [Decision Letter · Decision Letter 1]

30 Apr 2024

Construction and alidation of a severity prediction model for acute pancreatitis based on CT severity index: a retrospective case-control study,

PONE-D-24-02666R1

Dear Dr. Hu,

We’re pleased to inform you that your manuscript has been judged scientifically suitable for publication and will be formally accepted for publication once it meets all outstanding technical requirements.

Kind regards,

Jincheng Wang

Academic Editor

PLOS ONE

Additional Editor Comments (optional):

After revision, this manscript has been improved. I think this manuscript can be accepted for publication.

Reviewers' comments:

Reviewer's Responses to Questions

**Comments to the Author**

1. If the authors have adequately addressed your comments raised in a previous round of review and you feel that this manuscript is now acceptable for publication, you may indicate that here to bypass the “Comments to the Author” section, enter your conflict of interest statement in the “Confidential to Editor” section, and submit your "Accept" recommendation.

Reviewer #2: (No Response)

Reviewer #3: All comments have been addressed

2. Is the manuscript technically sound, and do the data support the conclusions?

Reviewer #2: Yes

Reviewer #3: Yes

3. Has the statistical analysis been performed appropriately and rigorously? 

Reviewer #2: Yes

Reviewer #3: Yes

4. Have the authors made all data underlying the findings in their manuscript fully available?

Reviewer #2: Yes

Reviewer #3: Yes

5. Is the manuscript presented in an intelligible fashion and written in standard English?

Reviewer #2: Yes

Reviewer #3: Yes

6. Review Comments to the Author

Reviewer #2: In this study, a model was constructed based on CTSI and blood urea nitrogen, neutrophil to lymphocyte ratio, d-dimer and ascites to predict the severity of acute pancreatitis, which has good clinical application value compared with other existing prediction models. However, the following problems still exist in this study.

1.The author collected the preliminary blood test results after the onset of the patient, but the serological indicators are affected by the length of the patient's disease course. In this study, the author did not indicate the length of the patient's disease course at the time of admission. This may affect subsequent research.

2. Serum calcium level is also considered to be a serological indicator that can accurately predict the severity of acute pancreatitis. Why was this indicator not included in this study?

Reviewer #3: I think the statistical analysis in this manuscript is reasonable and the data provided can support the conclusion. In addition, the language of the manuscript is concise and understandable.

7. PLOS authors have the option to publish the peer review history of their article (what does this mean?). If published, this will include your full peer review and any attached files.

Reviewer #2: No

Reviewer #3: No
